# An Overview of Rift Valley Fever Vaccine Development Strategies

**DOI:** 10.3390/vaccines10111794

**Published:** 2022-10-25

**Authors:** Paul Kato Kitandwe, Paul F. McKay, Pontiano Kaleebu, Robin J. Shattock

**Affiliations:** 1MRC/UVRI & LSHTM Uganda Research Unit, Entebbe 31410, Uganda; 2Department of Infectious Diseases, Division of Medicine, Imperial College London, Norfolk Place, London W2 1PG, UK

**Keywords:** Rift Valley fever virus, RVFV, Rift Valley fever vaccine, RVF

## Abstract

Rift Valley fever (RVF) is a mosquito-borne viral zoonosis that causes high fetal and neonatal mortality in ruminants and a mild to fatal hemorrhagic fever in humans. There are no licensed RVF vaccines for human use while for livestock, commercially available vaccines are all either live attenuated or inactivated and have undesirable characteristics. The live attenuated RVF vaccines are associated with teratogenicity and residual virulence in ruminants while the inactivated ones require multiple immunisations to induce and maintain protective immunity. Additionally, nearly all licensed RVF vaccines lack the differentiating infected from vaccinated animals (DIVA) property making them inappropriate for use in RVF nonendemic countries. To address these limitations, novel DIVA-compatible RVF vaccines with better safety and efficacy than the licensed ones are being developed, aided fundamentally by a better understanding of the molecular biology of the RVF virus and advancements in recombinant DNA technology. For some of these candidate RVF vaccines, sterilizing immunity has been demonstrated in the discovery/feasibility phase with minimal adverse effects. This review highlights the progress made to date in RVF vaccine research and development and discusses the outstanding research gaps.

## 1. Introduction

### 1.1. Rift Valley Fever Epidemiology

Rift Valley fever (RVF) is a mosquito-borne zoonotic viral disease caused by the Rift Valley fever virus (RVFV). The disease derives its name from the Rift Valley region of Kenya where it was first described in 1931 during an epidemic outbreak that was associated with high rates of abortion among pregnant ewes and acute deaths of newborn lambs [1]. The RVFV is transmitted from mosquitoes to animals or humans and from infected animals to humans [2]. Animal-to-animal and human-to-human transmission of the RVFV has not been demonstrated, but vertical transmission readily occurs in animals and has also been reported in humans [3,4]. Low-level circulation of the RVFV in livestock can occur without causing disease outbreaks. When RVF disease outbreaks occur, they are often associated with periods of heavy rainfall accompanied by flooding which creates conditions that facilitate the multiplication of competent mosquito vectors [5,6].

Rift Valley fever is a disease of public health importance. Although endemic to Africa, RVF has spread beyond this continent to other territories such as Madagascar, Comoros Islands, Saudi Arabia, and Yemen [7,8,9]. Due to the existence of several risk factors such as the presence of competent mosquito vectors, the wide range of susceptible domestic and wild animals, and climate change in RVF non-endemic areas, RVF has the potential to spread in these areas [10]. Rift Valley fever is considered a possible bioterror threat [11] and is listed as a notifiable disease by the world organisation for animal health-OIE [12]. The RVFV is also listed as a select overlap agent by the U.S. Department of Health and Human Services (HHS) and the U.S. Department of Agriculture (USDA) [13].

Rift Valley fever infects a variety of domestic and wild animals. Wild animals tend to have mild or inapparent infection whereas domestic animals are more susceptible to RVF disease [14]. Sheep and goats are the animals most susceptible to RVFV infection with mortality in newborn lambs and kids infected with this virus often exceeding 90% [15]. In pregnant sheep or goats, RVFV infection results in nearly 100% fetal mortality [16]. Rift Valley fever infection in older non-pregnant animals is often asymptomatic and abortion may be the only overt manifestation of the disease in a herd or flock [15]. In humans, the majority of those infected with the RVFV develop a mild to moderate form of RVF disease that is characterized by a subclinical non-fatal, febrile illness. However, a small percentage of RVF patients develop a severe ocular, encephalitic, or hemorrhagic form of the disease [17]. Ocular RVF is associated with eye lesions accompanied by blurred or decreased vision in one or both eyes which occasionally may result in permanent visual loss. Encephalitic RVF is characterized by intense headache, hallucinations, disorientation, vertigo, convulsions, excessive salivation, weakness, and partial paralysis, while hemorrhagic RVF causes symptoms such as jaundice resulting from liver damage, vomiting of blood, passing of blood in stool, ecchymoses, and nose or gum bleeding [16]. Approximately 50% of patients with the hemorrhagic form of RVF die, typically 3 to 6 days after symptom onset [18]. There is no specific treatment for RVF disease; mild cases are usually left to resolve on their own while general supportive therapy is given for the severe cases.

### 1.2. Molecular Biology of the Rift Valley Fever Virus

The RVFV is a Phlebovirus belonging to the order *Bunyavirales* and family *Phenuiviridae* [19]. This virus has a tripartite single-stranded RNA genome consisting of a small (S), medium (M), and large (L) segment (Figure 1). The S segment encodes in an ambisense manner the virus nucleoprotein (N) in the negative-sense orientation and the nonstructural (NSs) protein in the positive-sense orientation. The N encapsidates the viral RNA to form the ribonucleocapsid, a requirement for transcription and RNA replication [20]. The NSs protein is a major virulence factor in RVFV infection. It promotes the degradation of double-stranded RNA-dependent protein kinase (PKR) [21,22] and transcription factor II human (TFIIH) p62 subunit [23] and antagonizes host gene transcription of interferon-β, a cytokine that is important in establishing an anti-viral state [24,25]. The M-segment encodes in a negative sense at least four nested proteins namely the surface glycoproteins n (Gn) and c (Gc), and two non-structural proteins namely a 14 kDa protein Nsm, and the 78-kDa protein. Synthesis of the M segment proteins involves leaky scanning of the ribosome at five initiation codons within the NSm region [7]. The 78 kDa protein is synthesized from the first initiation codon, NSm from the second initiation codon while the fourth initiation codon gives rise to Gn and Gc only [26]. The functions of the Nsm and the 78 kDa protein are not fully understood but the former has been implicated in viral pathogenesis by suppressing virus-induced apoptosis [27] and the latter in the transmission of the RVFV from mosquitoes to ruminants, with a possible role in the replication of the virus in the mosquito host [28]. The surface glycoproteins Gn and Gc form heterodimers on the surface of the RVF virion and are essential in virus attachment to initiate infection. Rift Valley fever virus infection occurs through the interaction of Gn and Gc with CD209 and CD209L receptors [29,30] followed by caveolin-1-mediated endocytosis with Gc adopting a class-II fusion protein fold to facilitate a pH-dependent merger of the host and virion membrane [31,32]. The L-segment encodes the viral RNA-dependent RNA polymerase (RdRP) which synthesizes both viral mRNA and genomic RNA [33].

The Rift Valley fever virus genome consists of three single-stranded RNA segments S (small), M (medium), and L (large). These segments are encapsidated by the nucleoprotein (N) into ribonucleoproteins which are associated with the viral polymerase (L). The virion surface is covered by heterodimers of Gn and Gc.

### 1.3. Immune Responses against Rift Valley Fever Virus

Whereas the true mechanistic correlates of immune protection against RVFV in humans or animals are not known [34], based on early studies, the generation of neutralizing antibodies against Gn and Gc has provided a good correlate of protection [35]. Thus, Gn and Gc have been the main antigen targets for the development of RVF vaccines [36]. Nucleoprotein N induces high levels of IgG and IgM antibodies in RVFV and other bunyavirus infections but these antibodies are not neutralizing [37]. Anti-N immune responses however appear to play a role in protection against RVFV with this protein being a target for antibody-dependent cell-mediated cytotoxicity (ADCC) and complement-dependent cytotoxicity (CDC) as well as a potent human CD8+ T cell antigen [38,39]. In African green monkeys challenged with RVFV, the early proliferation of CD4+ and CD8+ T cells and the expression of Th1 cytokines were associated with non-lethal outcomes [40]. In humans, a higher concentration of IL-10 a cytokine that suppresses Th1 responses was associated with fatal cases compared to non-fatal cases in humans [41].

## 2. Rift Valley Fever Vaccine Development

Currently, there is no licensed RVF vaccine for human use although two inactivated vaccines namely NDBR-103 and TSI-GSD-200, and one live attenuated vaccine MP-12 have been tested and had limited use in humans [42]. In contrast, a handful of RVF vaccines that are based on conventionally live attenuated and inactivated vaccine platforms have been licensed for veterinary use but these have been associated with suboptimal safety and potency, respectively [43]. Additionally, these vaccines lack the property of differentiating infected from vaccinated animals (DIVA), making their universal adoption in RVF non-endemic countries difficult. As a result of these limitations, considerable research has been ongoing to develop safer RVF vaccines for both animal and human use (Table 1). This review provides an update on the research and development of RVF vaccines to date, highlighting the progress made and the outstanding research gaps.

### 2.1. Conventional Live Attenuated RVF Vaccines

#### 2.1.1. Smithburn

The first and one of the most widely used livestock RVF vaccine is the Smithburn vaccine developed in 1949 by attenuation of the RVF Entebbe strain virus through serial passage in mouse brain [44]. The RVFV used to make this vaccine had been isolated in 1944 from an *Eretmapodites* species mosquito captured in Uganda [45]. In calves vaccinated with the Smithburn vaccine, neutralizing antibody titers reached ≥1:64 by day 14 and ≥1:192 by day 28 post-vaccination. Vaccinated animals did not show any clinical signs of RVF and viremia could not be detected after challenge with virulent RVFV [46]. Despite its relatively low cost and ability to induce long-lasting immunity after a single administration, numerous safety concerns associated with the use of the Smithburn vaccine have been raised. The vaccine causes high abortion rates in pregnant animals and can cause harmful changes in internal organs with the attenuated virus propagating itself inside hepatic cells such as a natural RVFV infection. Alpacas immunized with the Smithburn vaccine resulted in meningoencephalitis and death in 3 out of 4 animals [47]. The vaccine caused abortions in 28% of immunized European cows and resulted in utero transmission of the vaccine virus [48] while goats developed severe deleterious pathological changes in the liver with the vaccine virus disseminating to numerous organs and tissues [49]. Due to these safety issues, the Smithburn vaccine is not recommended for use in countries where RVFV has not yet spread even though it is still used in some countries where this virus is endemic such as South Africa and Kenya [50].

#### 2.1.2. MP-12

The MP-12 RVF vaccine was developed in 1985 as a live-attenuated vaccine candidate targeted for both human and animal use. It was generated via 12 serial plaque passages of the pathogenic RVFV strain ZH548 in MRC-5 cells in the presence of the mutagen 5-fluorouracil [51]. The attenuation process resulted in one, three, and seven amino acid substitutions in the S, M, and L segments, respectively, of the RVFV genome [52]. The immunogenicity and safety of the MP-12 vaccine has been studied extensively in several animal models. No viremia was seen after vaccination of ewes with MP-12 and after post-vaccination challenge with virulent RVFV. Lambs less than 7 days old, produced neutralizing antibodies 5–7 days after vaccination, with protection against virulent RVFV challenge being achieved within 2 weeks at a dose of 5 × 10^3^ plaque-forming units (PFU). Other than brief pyrexia and a transient low titer viremia, no other untoward effects were seen [53]. In an independent evaluation study aimed at rigorously evaluating the virulence and safety of MP-12, calves and lambs vaccinated with this vaccine at a higher dose of 10^6^ PFU as opposed to the manufacturer’s recommended dose of 10^3^ PFU did not develop detectable viremia. By day 6 and 14, all lambs and calves, respectively, had developed protective antibody titers (PRNT_80_ ≥ 1:40). No pathological lesions associated with RVFV infection or viral RNA were detected in lamb tissues although there was detectable viral RNA in the tissues from 5 out of 10 calves, with liver necrosis in 3 of them [54]. Neutralizing antibodies of (PRNT_70_ ≥ 40) were still present 24 months after MP-12 vaccination with no evidence of vector transmission of the vaccine virus [55]. Pregnant cows vaccinated with MP-12 developed transient postvaccination viremia titer ≥ 2.5 log10 PFU/mL of serum but their calves were RVFV negative at birth, and the vaccine virus was not shed in their milk [56]. In Rhesus monkeys, MP-12 inoculated intravenously produced transient, low-titer viremia and minimal serum enzyme elevations. In addition, neurovirulence was observed but this caused only mild residual lesions with a severity similar to that caused by the 17D yellow fever vaccine [57]. All monkeys immunized by the mucosal route developed PRNT_80_ ≥ 1:40 by day 21 after inoculation and were protected from an aerosolized or intravenous virulent virus challenge. Long-term (>6 years) protection of the MP-12 vaccine was also demonstrated in Rhesus monkeys [60,61].

In a phase 1 dose escalation and route comparison study in humans, a single dose of MP-12 was highly immunogenic with 93% of recipients developing anti-RVFV specific antibodies. Up to 82% of vaccinees remained seropositive (PRNT_80_ ≥ 1:20) for at least one year after vaccination [134]. In a phase 2 clinical trial, 95% and 100% of vaccines had achieved a PRNT_80_ and PRNT_50_ ≥ 1:20, respectively, 28 days after MP-12 vaccination. All the PRNT_80_ responders maintained a PRNT_80_ and PRNT_50_ ≥ 1:40 for at least 12 months postvaccination. Sequencing of RNA from MP-12 isolates found no reversions of amino acids to those of the parent virulent ZH548 virus. Five years after a single dose of the MP-12 vaccine, 89% of vaccinees maintained a PRNT_80_ ≥ 1:20. However, low viremia was detected in some of the vaccinees after double-blind passage on Vero cells during the first 14 days after vaccination [58].

Despite the effectiveness and generally good safety profile of the MP-12 vaccines observed in numerous studies, some serious adverse events have also been reported. The vaccine was reported to cause multifocal random areas of hepatocellular degeneration and necrosis in calves [54] and teratogenic effects and abortions in sheep when administered in the first trimester [59].

### 2.2. Formalin Inactivated Vaccines

#### 2.2.1. NDBR-103

Randall et al. developed a formalin-inactivated RVF vaccine from monkey kidney cell cultures infected with the pantropic Entebbe strain of the RVFV that had undergone 176 intraperitoneal or intravenous passages in mice. This vaccine was safe with only two mild reactions out of more than 1000 immunized persons. It was also immunogenic with 40 out of 43 persons that received a three-dose primary series developing neutralizing antibodies of log 2 or more [62]. A lyophilized version of this vaccine NDBR-103 provided an adequate antibody response of log 1.7 or greater in 84% of persons that received a three-dose primary series and protection of up to 20 months after boosting [63]. Maximum antibody responses of the NDBR-103 were seen at six weeks following a three-dose primary vaccination series. Six months, one year, and two years later 88%, 91%, and 74% of the vaccines had antibodies detectable by PRNT ≥ 10, respectively [64].

#### 2.2.2. TSI-GSD-200

The NDBR-103 vaccine had a relatively good immune response, but it was produced using a vaccine seed virus inoculum that was composed of infectious mouse serum and primary monkey kidney cells as the substrate raising some safety concerns. Consequently, a similar vaccine TSI-GSD-200 manufactured under rigorous safety regulations was developed using well-characterized diploid fetal rhesus lung cells. Reactions to TSI-GSD-200 vaccination were mild causing limited local reactions and no fever, systematic reactions, or significant clinical laboratory abnormalities. Titers of 1:40 or more which are considered protective against RVFV infection were observed at least once in all 1.0 mL doses and in 30/31 of the 0.3 mL dose recipients. However, significant variability was observed in the geometric mean titer evoked by the different vaccine lots [65]. Peak geometric mean PRNT titers registered on day 42 ranged from 48 to 436 for the 0.1 and 1.0 mL dose, respectively. However, a sharp decline in antibody titers was observed by day 84 and six months after vaccination with apparently protective antibody titers being seen only in groups that received the higher doses (0.5–1.0 mL) of the most antigenic lot of the vaccine [66]. A secondary booster shot produced peak antibody titers that occurred earlier (day 10) and these were significantly greater than those observed after the 3-shot primary series but still, a marked decline in antibody titers was also evident by day 180 [67]. Protective immune responses (PRNT_80_ ≥ 1:40) to the 3-dose primary vaccination series with TSI-GSD-200 were reported to occur in 90% of vaccinees. In these vaccine recipients, there was a 50% probability that the protective immune responses would be maintained for about 8 years after a booster shot. In contrast, in the 3-dose primary series non-responders, the protective immune responses were predicted to last for only 204 days after their induction following boosting [68]. While the TSI-GSD-200 vaccine has been demonstrated to be safe, the requirement for multiple shots to induce and maintain protective immunity is a major limitation. Interestingly, RVFV-specific T-cell responses to formalin-inactivated RVF vaccines were reported to be long-lasting, being detectable up to 24 years after vaccination in some vaccine recipients [34].

### 2.3. Genetically Modified Live Attenuated RVF Vaccines

#### 2.3.1. Clone 13

The Clone 13 vaccine is another widely used live attenuated RVF vaccine that was developed using the RVFV strain 74HB59 which had a 69% natural deletion in the pathogenic NSs gene [69]. This strain was obtained from a nonfatal human case of RVFV infection during the 1974 RVF outbreak in the Central African Republic [69]. The Clone 13 vaccine was reported not to induce RVF symptoms such as abortion in pregnant ewes, teratogenicity in their offspring, or pyrexia. It also prevented clinical RVF following virulent challenge at different stages of pregnancy [70]. In goats and sheep, the frequency of adverse events in Clone 13 vaccinated animals was similar to those that received placebo immunizations. Seroconversion rates as measured by virus neutralization test and IgG and IgM ELISA were about 70% and these persisted for about a year [71]. A blinded randomized controlled field trial conducted in Kenya in sheep, goats, and cattle showed that Clone 13 did not cause any of the RVFV infection-associated symptoms. Neutralizing IgG antibodies were detected in over 90% of the immunized sheep and goats although only two-thirds of cattle developed them [72]. A vaccine trial conducted on goats, sheep, and cattle in Tanzania reported no adverse reactions, abortions, or deaths associated with Clone 13 vaccination. However, only 5 out of 300 vaccinated animals were IgM positive and IgG responses in cattle were substantially lower than those of goats and sheep [73].

A thermostabilized version of the Clone 13 was made by passing this naturally attenuated live virus through three cycles of heating at 56 °C and selecting thermostable clones. This vaccine named CL13T was found to be safe in camels and did not induce abortions or teratogenic effects. A single dose of CL13T stimulated a strong and long-lasting neutralizing antibody response that lasted for up to 12 months [74]. At room temperature (24 °C), the infectivity titers of CL13T were reported to decrease by only 0.4 logs after 3 days of storage and remained stable for at least 15 days [75].

Whereas nearly all studies have shown that the Clone 13 RVF vaccine is highly immunogenic and safe, at least one study reported that this vaccine can cross the ovine placental barrier and cause fetal infections, stillbirths, and malformations of the central nervous or skeletal system when administered in an overdose to pregnant ewes in their first trimester [76].

#### 2.3.2. ArMP-12ΔNSm21/384

To improve the safety profile of the MP-12 vaccine, mutations mainly targeting the pathogenic NSs and Nsm genes were introduced into this attenuated RVFV using reverse genetics. The safety and immunogenicity of a live attenuated vaccine arMP-12ΔNSm21/384 developed using this approach has been extensively evaluated in animals. Pregnant ewes immunized with arMP-12ΔNSm21/384 during the early (G1) (<35 days) and late (G2) (>35 days) stages of pregnancy were all positive for neutralizing antibodies two weeks post-vaccination with average titers of 2 and 2.5 logs for the G1 and G2 stages, respectively. Whereas none of the ewes aborted during pregnancy and all ewes vaccinated during the late G2 stage gave birth to healthy lambs, some of the ewes vaccinated during the G1 stage gave birth to lambs with deformities [77]. When tested in 4–6 months old calves, no significant adverse clinical events were observed attributed to arMP-12ΔNSm21/384. This vaccine was immunogenic at doses of 1 × 10^1^ through 1 × 10^7^ PFU with doses of 1 × 10^4^ or 1 × 10^5^ PFU stimulating presumed protective PRNT_80_ responses for at least 91 days after vaccination [78]. This vaccine was further tested in crossbreed ewes at 30–50 days of gestation. Neutralizing antibody was first detected in 3 of 4 animals by day 5 post-inoculation and all four animals had PRNT_80_ titers of ≥1:20 on Day 6. A 1 × 10^3^ PFU vaccine dose stimulated a PRNT_80_ response comparable to doses of up to 1 × 10^5^ PFU [79]. A single vaccination with arMP-12ΔNSm21/384 fully protected sheep from viremia and fever when challenged with the virulent RVFV ZH501 strain four weeks after vaccination. In vitro, arMP-12ΔNSm21/384 induced IFN-γ secretion by peripheral blood mononuclear cells (PBMCs) on day 14 was significantly higher than that observed in unimmunized controls suggesting a role of cell-mediated immunity in protection from RVFV infection [80]. Goats vaccinated intramuscularly with the arMP-12ΔNSm21/384 vaccine developed neutralizing antibodies that were detectable in some of them by day 5 and in all of them by day 7. By day 28, all goats had developed protective PRNT_80_ titers of ≥1:40. All goats remained healthy, and viremia was not detected in any of them after vaccination [81]. Evaluation of the arMP-12ΔNSm21/384 in sheep, goats, and cattle showed that this vaccine did not spread from vaccinated to unvaccinated animals and that there was no evidence of reversion to virulence in sheep. Even with an overdose, the vaccine did not cause any adverse effects. However, in calves, the antibody response was delayed, and the titers were lower than those of sheep and goats [82].

The efficacy of post-exposure vaccination with MP-12 viruses lacking the NSs gene has also been investigated. Vaccination of mice with this NSs deleted RVFV resulted in 30% survival when administered within 30 min of subcutaneous virulent RVFV challenge while the parental MP-12 virus conferred no protection by post-exposure vaccination [135]. In hamsters, a more extended post-exposure prophylaxis window was observed with a survival rate of 80%, 70%, and 30% when vaccination was delayed for 8, 12, and 24 h after virulent RVFV exposure [83]. This protection was postulated to be due to the induction of host innate immune responses induced by the replicating vaccine as a result of the functional inactivation of the truncated NSs protein.

#### 2.3.3. RMP-12-GM50

A recombinant MP-12 vaccine (rMP-12-GM50) having 584 silent mutations in all three RVFV genome segments was successfully rescued via the reverse genetics system. A total of 326, 185, and 73 mutations were introduced in the L, M, and S segments, respectively, at 50 nucleotide intervals with each silent mutation designed to not disturb codon-pair bias in humans. Vaccination of mice with 5 × 10^5^ PFU of this vaccine via the intramuscular route provided 100% protective efficacy after virulent RVFV challenge as did the parental MP-12 vaccine. Vaccination via the subcutaneous route resulted in a lower protective efficacy of only 50% and 60% for rMP-12GM50 and parental MP-12, respectively. A reduction in viral replication was however observed for the rMP-12-GM50 vaccine which was also reflected in lower PRNT_80_ titers [84].

#### 2.3.4. R566

To further improve the safety of the Clone 13 and MP-12 vaccines, a live attenuated recombinant RVFV vaccine R566 that combined the S segment of Clone 13 and the L and M segments of MP-12 was generated via reverse genetics. The efficacy of R566 was compared with that of a non-spreading RVFV Gn replicon (NSR-Gn) vaccine using Clone 13 as a control. Groups of eight lambs were vaccinated once and challenged after three weeks. All mock-vaccinated and challenged lambs developed a high fever and viremia three of which died. Two lambs vaccinated with the R566 vaccine developed a mild fever following virulent RVFV challenge and this was associated with low levels of viremia. None of the lambs vaccinated with the NSR-Gn and Clone 13 vaccine developed viremia or clinical signs of infection after virulent RVFV challenge [85].

#### 2.3.5. RRVF-ΔNSs:GFP-ΔNSm and ΔNSs-ΔNSm rRVFV

Bird et al. developed a reverse genetics-derived recombinant enhanced green fluorescent protein (eGFP)-tagged RVFV containing complete NSs and NSm deletions. This live-attenuated virus vaccine rescued from the RVFV ZH501 strain sequences conferred complete protection from both clinical illness and lethality in 100% of the vaccinated rats challenged with a lethal dose of the virulent RVFV. High-level protective immunity was induced by a single vaccine dose with 37 out of 40 total vaccinated rats developing sterilizing immunity [86]. The same vaccine without the eGFP tag (ΔNSs-ΔNSm rRVFV) was evaluated in pregnant sheep. All (20/20) pregnant ewes vaccinated on day 42 when the risk of RVFV vaccine-related teratogenesis is highest, progressed to full-term delivery producing lambs without any congenital abnormalities. Of these, all (9/9) pregnant ewes that were challenged with virulent RVFV obtained sterilizing immunity and delivered healthy lambs [87]. In the common marmoset (*Callithrix jacchus*), a non-human primate (NHP) model, a single vaccine dose of this vaccine led to the development of robust antibody responses with no vaccine-induced adverse reactions, signs of RVFV infection, or infectious virus. All vaccinated animals subsequently challenged with virulent RVFV were protected against viremia and liver disease [88]. Very low mosquito dissemination and impaired transmission of the human vaccine candidate version of this vaccine (DDvax) was observed in vaccinated livestock when compared to the MP-12 vaccine and the wild-type parental virus ZH501. Only 1 out of 260 mosquitoes produced vaccine plaque from their saliva after a high titer challenge suggesting that the transmission and dissemination of this vaccine by mosquitoes from vaccinated individuals during an epidemic was highly unlikely [89].

#### 2.3.6. Four Segmented RVFV 4S

Wichgers Schreur et al. developed a novel live attenuated virus vaccine RVFV 4S which had the L segment, the S segment with the NSs gene removed, and the M segment split into two segments encoding either the Gn or Gc [90]. This vaccine-induced sterile immunity in lambs after a single vaccination, when administrated via the intramuscular route [91]. The virus did not cause encephalitis after intranasal inoculation of mice, and in pregnant ewes, did not cause viremia or cross the ovine placental barrier [92]. The RVFV-4s candidate vaccine for veterinary use (vRVFV-4s) did not disseminate in vaccinated animals, was not shed or spread to the environment, and did not revert to virulence. A single vaccination of lambs, goat kids, and calves with this vaccine induced protective immunity against a homologous challenge and provided full protection against a genetically distinct RVFV strain [93]. In pregnant ewes, a single vaccination of vRVFV-4s provided complete protection from vertical transmission and abortion [94].

### 2.4. DNA Vaccines

Spik et al. used the pWRG7077 plasmid to encode the RVFV M segment starting at the second or fourth methionine which code for Gn and Gc with (RVFV + NSm DNA) or without (RVFV-NSm DNA) the Nsm protein, respectively. Following gold particle precipitation of the pDNA, these vaccines were administered to mice using a gene gun in three separate administrations. Whereas the two vaccine constructs had similar in vitro antigen expression levels, RVFV-NSm DNA was highly immunogenic and protective compared to RVFV + NSm DNA which was neither immunogenic nor protective [95].

A DNA vaccine encoding the nucleoprotein N (RVFV cDNA N) and another encoding Gn and Gc (RVFV cDNA GnGc) using the pcDNA3.1/V5-His^®^ TOPO vector (Invitrogen, Waltham, MA, USA) protected four out of eight, and five out of eight mice, respectively, from clinical signs of RVFV infection after virulent virus challenge. These vaccines which were delivered by gene gun in four different administrations after precipitation of the DNA on gold beads produced PRNT_50_ tires of <25 and 25–75 for the nucleoprotein N and the Gn-Gc DNA vaccines, respectively [96].

A DNA plasmid PTR600 expressing RVFV Gn coupled to 3 copies of the complement protein C3d as a molecular adjuvant (Gn-cd3 DNA) elicited high titer neutralizing antibodies in mice that were comparable to those elicited by the MP-12 vaccine. The vaccine protected vaccinated mice against lethal RVFV challenge and completely prevented weight loss and morbidity in these mice. Passive transfer of antisera from vaccinated mice into naive mice protected them as effectively as sera from mice immunized with MP-12 [97].

In a transgenic mouse model with impaired interferon type I response (IFNAR−/−), dose-dependent protection in animals immunized with a DNA vaccine that utilised the pDNA vector pCMV (Clontech) to encode both mature RVFV Gn and Gc (pCMV-M4) was achieved. In contrast, only partial protection was achieved in mice vaccinated with either pDNA expressing N (pCMV-N) or a combination of both vaccines (pCMV-M4 + pCMV-N) [98]. The same vector encoding Gn and Gc (pCMV-GnGc) administered in a homologous DNA or heterologous DNA+ Modified Vaccinia virus Ankara (MVA) prime-boost regimen reduced clinical signs and viremia in adult sheep challenged 12 weeks after the last immunization. Protection was associated with the production of neutralizing antibodies despite the lack of reactivity of the anti-Gn and anti-Gc specific ELISA before challenge [99].

Chrun et al. evaluated in sheep the protective immunity induced by DNA vaccines encoding the extracellular portion of Gn targeted to the dendritic cell makers CD11c (pscCD11c-eGn) or DEC205 (pscDEC-eGn) or untargeted (peGn). The untargeted DNA was more potent at inducing IgG responses compared to the targeted DNA and conferred significant clinical and virological protection upon infectious RVFV challenge. Protection was associated with the anti-eGn IgG antibodies, rather than the T-cell response but surprisingly, these antibodies were not neutralizing in vitro. Other mechanisms of protection such as complement-dependent inhibition, antibody-dependent cellular cytotoxicity (ADCC), or differences in viral receptor engagement between the in-vitro assay and in vivo conditions were postulated to be at play [100].

Gonzalez-Valdivieso et al. evaluated DNA vaccines encoding RVFV Gn fused to elastin-like recombinamers (Gn-ELRs). Elastin-like recombinamers are a type of elastin-like polypeptides (ELPs) biosynthesized through genetic engineering. Elastin-like polypeptides are polymers formed by short pentapeptide VPGXG repeats found in a natural elastin sequence where X can be any amino acid except proline. Even though glutamic acid or valine-rich fusion proteins had more in vitro expression compared to the pDNA expressing non-fused Gn, when evaluated in mice, they and all other Gn-ELR fusion constructs had lesser efficacy [101].

The plasmid phMGFP vector (Promega) that encodes a Monster Green^®^ Fluorescent Protein (mGFP) was cloned with full-length Gn and Gc genes of RVFV strain 1974-VNIIVVi to produce phRVF/Gn and phRVF/Gc DNA vaccines, respectively. This vector which was also cloned with fusion protein F sequences of human parainfluenza (HPIV-1) in order to provide correct post-translational modifications was encapsulated in biodegradable alginate (ALG)/poly-L-lysine (PLL) microcapsules. Mice immunised with a combination of phRVF/Gn and phRVF/Gc vaccines developed higher virus-neutralizing antibodies titers ranging from 1:16–1:32 compared to the free DNA plasmids whose titers were from 1:4–1:8 [102].

### 2.5. Virus Vectored Vaccines

#### 2.5.1. Lumpy Skin Disease Virus (rLSDV-RVFV)

A lumpy skin disease virus (LSDV) vectored recombinant vaccine expressing RVFV Gn and Gc (rLSDV-RVFV) induced neutralizing antibodies in sheep and mice and fully protected them from virulent RVFV challenge. In contrast, mice vaccinated with a DNA vaccine expressing these antigens did not seroconvert and only 20% survived virulent RVFV challenge. Boosting with the rLSDV-RVFV instead of the DNA vaccine increased the survival rate to 40% [103]. In cattle, none of the animals vaccinated with the rLSDV-RVFV showed clinical symptoms typical of RVFV infection after virulent RVFV challenge however, the virus was detected in two of the five animals, but with significantly lower viremia compared to the mock-vaccinated group. Neutralizing antibodies were present in three of the five calves after vaccination, which increased significantly by day 6 after challenge [104].

#### 2.5.2. Complex Adenovirus Vector (CAdax-RVF)

A nonreplicating complex adenovirus (CAdVax) vector was used to deliver RVFV Gn and Gc genes. In the absence of anti-vector antibodies, all mice (8/8) survived lethal RVFV challenge including those that were not boosted. Efficacy was low in mice that had anti-vector immune responses at 25% for those that received a low dose of the vaccine (2 × 10^6^ PFU) and 75% for those that received a high dose (2 × 10^8^ PFU), suggesting that pre-existing immunity to Ad5 lowers efficacy which can be overcome by increasing the vaccine dose [105].

#### 2.5.3. New Castle Disease Virus (NDFL-GnGC)

A homologous prime-boost vaccination of mice via the intramuscular route with a recombinant Newcastle disease virus (NDV) expressing RVFV Gn and Gc (NDFL-GnGc) protected them from lethal RVFV challenge. Whereas no clinical signs were observed in 50% of the mice, low levels of viral RNA (1 × 10^3^ RNA copies per ~5 mg) were detected in the liver at the end of the experiment although they did not seroconvert for antibodies against the N protein. A single vaccination of lambs with NDFL-GnGc induced RVFV-neutralizing antibodies that were significantly boosted after a second dose [106].

#### 2.5.4. Replication-Competent Vaccinia Virus (vCOGnGc and vCOGnGcγ)

Two recombinant RVF vaccines expressing RVFV Gn and Gc were developed using the replication-competent vaccinia virus (VACV) as a vector. The vaccines were attenuated by the deletion of a VACV gene encoding an IFN-γ binding protein and insertional inactivation of the thymidine kinase gene. One of the vaccines also expressed the human IFN-γ gene to enhance safety. Both vaccines were extremely safe and efficacious with survival from virulent RVFV challenge reaching 90% in mice that were double vaccinated using the vaccine lacking the IFN-γ gene. Single vaccination achieved much lower survival rates of 10% and 50% for the vaccine with and without the IFN-γ gene, respectively [107]. Baboons immunized with these recombinant vaccines all developed protective RVFV antibody titers although those that received the vaccine containing the IFN-γ gene had lower titers. The reduced immunogenicity and efficacy of the vaccine containing the human IFN-γ gene were postulated to be due to reduced replication caused by the attenuating effects of this gene on the recombinant VACVs.

#### 2.5.5. Replication-Deficient Chimpanzee Adenovirus (ChAdOx1-GnGc)

Warimwe et al. evaluated a replication-deficient chimpanzee adenovirus vector, ChAdOx1, encoding MP-12 strain Gn and Gc (ChAdOx1-GnGc) in comparison to a replication-deficient human adenovirus type 5 vector encoding Gn and Gc (HAdV5-GnGc). A single immunization of mice with either of these vaccines conferred protection against virulent RVFV challenge eight weeks after immunization. Both vaccines elicited RVFV-neutralizing antibodies and a robust CD8^+^ T cell response. The neutralizing antibody response induced by ChAdOx1-GnGc was significantly enhanced by the adjuvants Matrix-M^TM^ and to a lesser extent AddaVax^TM^ [108]. A single-dose immunization with the ChAdOx1-GnGc vaccine elicited high-titer RVFV-neutralizing antibodies and prevented viremia in virulent RVFV-challenged sheep, goats, and cattle. In dromedary camels, ChAdOx1-GnGc induced lower RVFV-neutralizing antibodies compared to sheep, goats, and cattle but they were within the range associated with protection observed in these and other livestock [109]. In pregnant sheep and goats, ChAdOx1-GnGc was safe, elicited high titer RVFV-neutralizing antibodies, and protected these animals against viremia and fetal loss although this protection was not as robust for goats [110]. A ChAdOx1-GnGc thermostabilized by slow desiccation on glass fiber membranes in trehalose and sucrose was tested for immunogenicity in cattle after storage for 6 months at 25, 37, or 45 °C. This vaccine elicited comparable RVFV-neutralizing antibody titers to those elicited by the cold chain vaccine that was stored at −80 °C and these were within the range associated with protection against RVFV infection in cattle [111]. A phase 1 clinical trial of the ChAdOx1 RVF vaccine is being conducted in the UK (NCT04754776) and Uganda (NCT04672824). The safety of the ChAdOx1vector in humans may be of concern considering the rare cases of Vaccine-induced Immune Thrombotic Thrombocytopenia (VITT) that have been reported in individuals immunized with the Oxford/AstraZeneca ChAdOx1-S COVID-19 vaccine which uses this vector [136,137].

#### 2.5.6. MVA Vectored (rMVA-Gn/Gc and rMVA-N)

Recombinant pDNA and Modified Vaccinia Ankara virus (rMVA) vectored vaccines encoding either RVFV Gn and Gc (rMVAGn/Gc) or the nucleoprotein N (rMVA-N) were evaluated for immunogenicity and protective efficacy in mice. A single dose of the rMVA-Gn/Gc vaccine-induced low-level RVFV neutralizing antibodies and glycoprotein-specific CD8+ T-cell responses in mice, preventing them from viremia or RVF symptoms following pathogenic RVFV 56/74 strain challenge. This protection was better than that of the DNA-Gn/Gc vaccine alone or a heterologous prime-boost vaccination schedule of DNA-Gn/Gc followed by rMVAGn/Gc. Whereas the rMVA-N vaccine induced high titer antibodies against the RVFV nucleoprotein N, it only conferred partial protection to virulent RVFV challenge whether in homologous or heterologous prime-boost schedules with the corresponding recombinant DNA vaccine [112]. Prime-boost subcutaneous immunization of adult sheep with homologous DNA or heterologous DNA/MVA encoding RVFV Gn and Gc induced a rapid in vitro neutralizing antibody response, IFNγ production, and reduced viremia upon virulent RVFV challenge. However, homologous MVA prime-boost vaccination showed increased viremia upon virulent RVFV challenge correlating with the absence of detectable neutralizing antibodies, despite induction of cellular responses after the last immunization. Interestingly, a faster induction of neutralizing antibodies and IFNγ production after challenge in this group compared to the mock vaccinated group was observed, which was indicative of a primed immune response [99]. In lambs, a single subcutaneous dose of the rMVA-Gn/Gc did not confer full protection against virulent RVFV challenge but delayed the onset and reduced the severity and duration of illness [113]. In the absence of in vitro neutralizing antibodies, protection of the rMVAGn/Gc was found to be mediated by the activation of cellular responses mainly directed against Gc epitopes [114].

#### 2.5.7. Bivalent MVA Vectored (MVA-GnGc-VP2, MVA-GnGc-NS1, and MVA-GnGc-NS1-Nt)

A bivalent MVA vectored vaccine expressing bluetongue virus (BTV) proteins NS1 and VP2 as well as RVFV Gn and Gc induced adaptive immune responses and protected both mice and sheep from BTV and RVF. Mice immunized with MVA-GnGc-NS1 did not develop any clinical signs or viremia after infection with virulent RVFV and were healthy throughout the experiment. In sheep, this vaccine lowered the mean rectal temperature and significantly reduced viremia following virulent RVFV challenge. The vaccine further protected sheep from liver damage as the levels of aspartate aminotransferase (AST), gamma-glutamyl transferase (GGT), and lactate dehydrogenase (LDH) did not significantly change after challenge with virulent RVFV [115].

#### 2.5.8. Equine Herpesvirus Type 1 (rH_Gn-Gc) and Capripoxvirus Recombinant Virus (rKS1/RVFV)

A vectored vaccine based on the equine herpesvirus type 1 (EHV-1) strain RaCH expressing RVFV Gn and Gc genes that were custom-synthesized after codon optimization induced protective titers that reached 1:320 at day 49 post-immunization [116]. Mice intraperitoneally vaccinated with a capripoxvirus (CPV) recombinant virus expressing RVFV Gn and Gc were fully protected from mortality after virulent RVFV challenge. Sheep subcutaneously vaccinated twice with this vaccine developed neutralizing antibodies and were protected against virulent RVFV as evidenced by the absence of viremia following virulent RVFV challenge [117].

#### 2.5.9. Rabies Virus Vector (rSRV9-eGn)

Zhang et al. developed an inactivated recombinant RVFV and rabies virus vaccine candidate encoding the RVFV Gn ectodomain. After combination with poly (I:C) and ISA 201 VG adjuvant, this vaccine-induced RVFV-specific IgG antibodies and cellular immune responses that led to the stimulation of IFN-γ, IL-4, and effector memory T cells. However, it failed to induce neutralizing antibodies, and this was postulated to be due to changes in the spatial conformation of eGn during recombinant virus packaging [118].

### 2.6. Subunit Vaccines

A recombinant subunit vaccine Gne-S3 containing the Gn ectodomain and adjuvanted in Stimune water-in-oil adjuvant (Prionics, Lelystad, The Netherlands) resulted in a neutralizing antibody response within three weeks after a single dose vaccination and protected lambs challenged with virulent RVFV from viremia, pyrexia, liver damage, and mortality [119]. A recombinant subunit vaccine composed of purified baculovirus-expressed Gn ectodomain (RVFV Gne) and Gc proteins (RVFV Gc) was formulated with a water-in-oil adjuvant montanide ISA25 (Seppic, France) and evaluated for immunogenicity in sheep. The primary dose of the vaccine-elicited putative protective PRNT_80_ titers ranging from 1:40–1:160 which were boosted to more than 1:1280 by the second dose. All animals tested positive for RVFV-neutralizing antibodies nearly a year after vaccination [120]. Following challenge with virulent Kenya-128B-15 RVFV strain, this vaccine conferred complete protection in all vaccinated sheep, as evidenced by the prevention of viremia, fever, mortality, and absence of RVFV-associated histopathological lesions [121]. Cattle vaccinated subcutaneously with either one or two doses of this subunit vaccine 35 days before virulent RVFV challenge were protected from viremia, fever, and RVFV-associated histopathological lesions [122].

Two emerging technologies—i.e., self-assembling multimeric protein scaffold particles (MPSPs) and “bacterial superglue” in which a covalent, intermolecular isopeptide bond is formed between the 13 amino acid “SpyTag” peptide and a small (12.3 kDa) “SpyCatcher” protein were used to make a subunit vaccine containing the head domain of RVFV Gn (Gn_head_ MPSPs). Lumazine synthase-based MPSPs reduced mortality in a lethal mouse model and protected lambs from viremia and clinical signs after immunization. When coupled to two other MPSPs (*Geobacillus stearothermophilus* E2 or a modified KDPG Aldolase) this subunit vaccine also provided full protection in lambs [123].

### 2.7. Virus Replicon Vaccines

Candidate RVF vaccines have also been made using virus replicon particles (VRPs). Virus replicon particles are genetically engineered viruses that lack the structural genes required for the successful infection of cells. Rift Valley fever virus VRPs are typically designed without the NSs and Nsm genes which are responsible for virulence and the Gn and Gc genes required for infection. The viral L and S genomes are transfected into packaging cell lines with the Gn and Gc genes being provided in trans, leading to the assembly of replicon-deficient VRPs. The VRPs can enter a target cell and undergo limited transcription and translation to synthesize encoded proteins but cannot produce infectious progeny due to the absence of the structural genes required for infection.

#### 2.7.1. Alphavirus Replicon Vectors

Alphavirus replicon vectors have been used to design RVF vaccines. In this vaccine format, a gene of interest replaces the viral structural genes under the control of a highly efficient internal sub-genomic promoter. The vector RNA is then packaged by providing the viral structural proteins in trans, resulting in replicon particles that can infect target cells and express the heterologous genes to high levels. Due to the lack of encoded homologous structural genes, these replicons are unable to produce progeny virions [138]. Alphavirus replicon vectors based on mosquito (AR86) and human (Girdwood) isolates of Sindbus virus were used to express RVFV Gn, Gc, and the NSm protein to create the pREP91-RVF (M) and Rgird-RVFV (M) vaccines, respectively. Although these vaccines induced relatively low virus neutralization titers of 1:4 to 1:16 after a two-dose vaccination in mice, all of them were protected against lethal RVFV challenge [124]. Gorchakov et al. compared Sindbis virus (SINV) and Venezuelan equine encephalitis virus (VEEV) vector genomes for their ability to express the RVFV Gn and induce a protective immune response against RVFV infection. Both the VEEV replicon (VEEVRepspGn) and SINV replicon (VEEVRepspGn) were capable of efficient Gn expression, however, after one immunization, only VEEV-specific, packaged replicons induced immune responses that protected mice against virulent RVFV ZH501 challenge possibly due to the higher levels of VEEV RNA replication in vivo and/or because of the stronger resistance of this virus’s replication in face of the autocrine action of IFN-α/β [125,139].

#### 2.7.2. Replication Deficient RVFV Replicons

To create a non-spreading RVFV Gn replicon (NSR-Gn) vaccine, baby hamster kidney (BHK)-GnGc cells were infected with a fowlpox virus expressing T7 polymerase and subsequently transfected with the pUC57 plasmids that encode the RVFV L and S-Gn genome segments, together with plasmid pCAGGS-M, which encodes the glycoprotein precursor. At the highest vaccination dose of 10^6.3^ TCID_50_/mL, this vaccine induced sterilizing immunity that protected all vaccinated lambs from clinical signs of RVFV infection and viremia following virulent RVFV challenge [126]. Single-cycle RVFV replicon particles (RVFV-VRPs) were made through the transfection of baby hamster kidney 21 (BHK-21) cells constitutively expressing T7 RNA polymerase (BSR-T7/5 cells) with plasmids encoding the RVFV wild-type L segment, S segment whose NSs gene had been replaced with the green fluorescent protein (GFP), and an expression vector carrying RVFV glycoprotein genes. A single-dose immunization of mice with 1.0 × 10^5^ or 1.0 × 10^4^ TCID_50_ of these RVFV VRPs conferred 100% protection against virulent RVFV challenge. Interestingly, robust protection from lethal challenge was observed very early with a survival rate of 60%, 80%, and 100% at 24, 48, and 96 h, respectively, after vaccination [127]. A single-cycle replicable MP-12 (scMP-12) vaccine was developed through co-transfection of BSR-T7/5 cells stably expressing T7 polymerase with plasmids, expressing the L, N, and Gn/Gc proteins, as well as the L RNA, S RNA encoding N protein and GFP and M RNA encoding a mutant envelope protein lacking an endoplasmic reticulum (ER) retrieval signal. This vaccine did not exhibit any neurovirulence after its intracranial inoculation in suckling mice in contrast to the MP-12 vaccine where all mice died. Vaccination of mice with 1.0 × 10^5^ PFU scMP-12 protected 90% of the animals from mortality after wild-type RVFV challenge and completely suppressed viremia and replication of the challenge virus in their livers and spleens [128]. Replacing the GFP gene of the scMP-12 vaccine with an NSs gene carrying the R16H/M250K mutation (scMP-12-mutNS) which lacked host transcription suppression function and moderately suppressed IFN-β transcription but retained PKR degradation function improved immunogenicity and protective efficacy of the scMP-12 [129]. Baby hamster kidney cells containing a recombinant fowlpox T7 polymerase were transfected with plasmids expressing RVFV L, S, and M segments resulting in replicon particles (RVFV RRP) that fully protected mice from mortality and clinical signs of infection following challenge with virulent RVFV 35/74 [130].

### 2.8. Virus-like Particles

Virus-like particles (VLPs) are artificial nanostructures that are composed of all or some of the proteins that form the viral capsid but lack the genomic material. These bio-nanoparticles are unable to replicate in the recipient but their ability to present viral antigens in their native conformation enables them to induce strong cellular and humoral immune responses [140]. Since VLPs lack genetic material, they are safer than live attenuated vaccines as there is no possibility of genetic reassortment and reversion to virulence. The immunogenicity and efficacy of the Ren-VLP vaccine made through transfection of 293T cells with expression plasmids for L, N, and M together with the reporter minireplicon construct vM-Ren were evaluated in mice. Three immunizations with 1 × 10^6^/dose of this vaccine produced high titers of virus-neutralizing antibodies protecting 11 out of 12 mice from virulent RVFV challenge. Whereas all but one mouse vaccinated with this VLP vaccine attained serum neutralizing titers of ≥250, similar titers were only found for half of the mice vaccinated with the lower dose of 1 × 10^5^ [131]. Chimeric VLPs containing RVFV Gn and Gc, nucleoprotein N, and the gag protein of Moloney murine leukemia virus (RVFV chimVLPs) provided long-lasting humoral and cellular immune responses in a mouse model following three vaccinations administered at 9-day intervals. PRNT_80_ levels of >1:640 were observed 161 days after the last vaccination while IL-2, IL-4, IL-5, and IFN-γ production was elicited by RVFV-specific antigen, consistent with both humoral (Th2) and cellular (Th1) responses. In mouse and rat lethal challenge models, high protection rates of 68% and 100%, respectively, were observed [132]. A VLP vaccine expressing RVFV Gn and Gc (RVFV GnGc VLP) was produced using the Drosophila insect cell expression system and formulated in the Stimune water-in-oil adjuvant. After two immunizations, this vaccine protected all mice challenged with virulent RVFV from any clinical signs of infection or mortality and provided 90% sterilizing immunity [141]. Mice were immunized with VLPs made by infecting suspended Sf9 insect cells with recombinant baculovirus expressing RVFV Gn, Gc, and N proteins. At 14 days from the second immunization, 60% and 100% neutralization at 1:74 and 1:32 serum dilution, respectively, was observed using a RVFV pseudovirus neutralization assay. High levels of the IL-4 and IFN-γ cytokines were also induced by CD4^+^ and CD8^+^ T cells pointing to a balanced Th1/Th2 immune response in mice [133].

## 3. Outstanding Gaps and Future Directions

To date, there are a handful of licensed RVF vaccines all of which are targeted for veterinary use. These vaccines which are based on either the classical live-attenuated virus, naturally attenuated live virus or inactivated virus platform, are however not fully licensed in RVF non-endemic areas as a result of having undesirable attributes [43]. Formalin-inactivated RVF vaccines require multiple doses to induce and maintain protective immunity making them unfavorable for use in RVF-endemic countries. In contrast, the most widely used live attenuated vaccine Smithburn even though efficacious has been associated with abortions, fetal malformations in pregnant animals, RVFV-associated pathologies, and mortality in some cases [47,48,49]. The Smithburn vaccine also carries a risk of genetic reassortment in nature with wild-type RVFV and reversion to virulence [7]. The other licensed naturally attenuated live virus vaccine Clone 13 has a much better safety profile compared to the Smithburn although there was a report that it can also cause fetal malformations when administered to pregnant ewes in an overdose during the first trimester [76]. In addition to the above limitations, the currently licensed RVF vaccines apart from Clone 13 are not DIVA compatible making them unsuitable for use in RVF non-endemic countries.

As a result of the shortcomings of the currently licensed veterinary and human RVF vaccines, significant research efforts have been ongoing to develop safer vaccine candidates. Vaccine development efforts have been greatly aided by a better understanding of the molecular biology of the RVFV, the protective immune responses that need to be induced, together with advancements in DNA technology. Despite their suboptimal safety, live attenuated vaccines induce robust protective immune responses after a single dose resulting in high vaccine efficacy. The use of reverse genetics to remove or mutate the genes responsible for virulence in wild-type and conventionally attenuated live RVF viruses has thus been a major strategy for the development of safer live attenuated RVF vaccines. These genetically modified live attenuated vaccines have been demonstrated to induce robust protective immune responses that are often similar to those induced by the Smithburn vaccine but with significantly milder and fewer adverse effects. Plasmid DNA vectors encoding either RVFV Gn or its ectodomain or both Gn and Gc have been shown to induce protective immune responses in mice and sheep. Unfortunately, DNA vaccines usually require special delivery methods such as gene guns as well as multiple vaccine administrations to induce and maintain protective immune responses. Novel technologies such as the use of pharmacokinetic enhancers such as elastin-like re-combinamers and microencapsulation are being explored to enhance the immunogenicity of DNA vaccines however, these are still in their infancy stages of development. Other vaccine platforms such as virus vectored, recombinant subunit proteins, replicon deficient virus replicons, and VLPs have exhibited superior safety profiles than the currently licensed live attenuated vaccines but with comparable immunogenicity and efficacy. The advancement of these candidate RVF vaccines into field trials with the goal of licensure is thus warranted.

Attempts to develop suitable human RVF vaccines have been made although progress has been slower compared to that of veterinary RVF vaccines partly due to the more stringent regulatory requirements for human vaccine licensure. Efficacy trials are a requirement for the licensure of human vaccines, but these may not be feasible for the development of vaccines against some outbreak diseases such as RVF whose occurrence is intermittent. Consequently, in 2002, the US Food and Drug Administration (FDA) introduced regulations that established the Animal Rule to grant marketing approval of drugs and biologicals when human efficacy studies are either not ethical or feasible, thereby opening up a new pathway for the licensure of human RVF vaccines [142]. Considering that human RVF epidemics are often preceded by outbreaks in livestock, a vaccination strategy focusing on animals when coupled with other measures such as vector control is still the best way to prevent human RVF outbreaks. Animal vaccination would also be beneficial irrespective of human transmission by preventing livestock deaths and averting significant economic losses, especially to pastoral communities.

A major limitation of most of the commercially available RVF vaccines is that they do not have the DIVA property making them unsuitable for use in RVF nonendemic. The only commercially available DIVA-compatible vaccine Clone 13 does not yet have commercially available diagnostic tests that can make use of this property [143]. Fortunately, the vast majority of new RVF vaccines being developed today are DIVA compatible making them appropriate for use in RVF non-endemic countries that are targeting control and eradication and for those exporting livestock products to RVF-free or RVF non-endemic countries. The development of these novel vaccines should be accompanied by the corresponding timely development of readily adoptable diagnostic assays.

Considering that RVF outbreaks are often sporadic, the manufacture and stockpiling of vaccines against this zoonotic disease can be expensive due to the typically short shelf life of vaccines. Furthermore, because RVF is endemic in Sub-Saharan Africa which has limited technical and financial capacity to manufacture vaccines, RVF vaccines should also be affordable and easy to manufacture. A novel vaccine platform that is suitable for rapid manufacture is messenger RNA (mRNA) which can be either in the conventional non-amplifying or self-amplifying format. The utility of mRNA in rapid vaccine production was demonstrated when the first two Coronavirus disease 2019 (COVID-19) vaccines which were approved and are among the most efficacious to date, were developed using this platform [144,145]. The cost of mRNA vaccines can potentially be lowered with the use of self-amplifying RNA (saRNA) which encode the mRNA sequence of the antigen of interest and that of an alphavirus RNA-dependent RNA polymerase (RdRP). The RdRP gene enables saRNA to replicate itself once delivered into the cell cytoplasm thereby facilitating the use of lower amounts of RNA compared to conventional mRNA vaccines [146]. Whereas no saRNA vaccine has yet been approved for use, several have undergone preclinical evaluation but only about five have undergone or are currently undergoing clinical evaluation [147,148].

Another factor to consider in the development of new RVF vaccines is thermostability. In many Sub-Saharan countries particularly in the rural areas, electricity supply is unavailable or unreliable making the maintenance of the cold chain in this largely topical region difficult [149]. Research and development of thermostable RVF vaccines is therefore important to enable the efficient distribution of RVF vaccines in this region. In this respect, thermostabilized versions of Clone 13 and ChAdOx1 RVF vaccines have been developed [74,111] with efforts ongoing to do the same for other vaccine platforms such as RNA [150].

## 4. Conclusions

Great advances have been made in the development of new RVF vaccines that have better safety and efficacy profiles than the commercially available live attenuated and inactivated vaccines, respectively. These vaccines are based on a variety of platforms including DNA, recombinant subunit proteins, VLPs, replication-deficient virus replicons, genetically attenuated live viruses, and virus vectors. For some of these candidate RVF vaccines, sterilizing immunity has been demonstrated in discovery/feasibility phase studies with minimal adverse effects. Although the majority of the RVF vaccines under development are targeted for veterinary use, a vaccine targeting both livestock and humans may be on the horizon with clinical trials of ChAdOX1 RVF, a vaccine that uses a vector that is used in a licensed COVID-19 vaccine underway in the UK and Uganda. As has been demonstrated with COVID-19, an ideal vaccine for an epidemic such as RVF does not only need to be safe and efficacious but also affordable and easy to produce, store and transport so that it can be efficiently distributed globally. More research should therefore be put towards the development of RVF vaccines and the infrastructure that will most appropriately address the current global vaccine production and distribution constraints.

## Figures and Tables

**Figure 1 vaccines-10-01794-f001:**
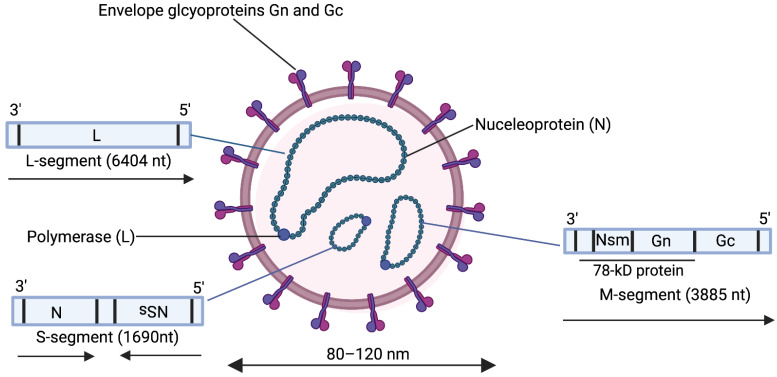
Structure and genomic organization of Rift Valley fever virus. Created with Biorender.com.

**Table 1 vaccines-10-01794-t001:** Rift Valley fever vaccine development strategies.

Vaccine Type	Vaccine Name	Vaccine Design	References
Conventional live attenuated	Smithburn	Neurotropic Smithburn Rift Valley fever virus (RVFV) strain attenuated through 102 serial passages in mouse brain	[44,45,46,47,48,49,50]
	MP-12	RVFV ZH548 strain plaque passaged 12 times in human fetal lung fibroblast cells (MRC-5) in the presence of the mutagen 5 fluorouracil	[51,52,53,54,55,56,57,58,59,60,61]
Inactivated	NDBR103	Formalin-inactivated RVFV Entebbe strain cultured in monkey kidney cells	[62,63,64]
	TSI-GSD-200	Formalin-inactivated RVFV Entebbe strain cultured in diploid fetal rhesus lung cells	[34,65,66,67,68]
Genetically modified live attenuated	Clone 13	Plaque purified naturally mutated RVFV 74HB59 strain having a 69% deletion in the nonstructural protein S (NSs) gene	[69,70,71,72,73,74,75,76]
	arMP-12ΔNSm21/384	Recombinant MP-12 virus with deletions in the NSm gene	[77,78,79,80,81,82,83]
	rMP-12-GM50	Recombinant MP-12 virus with a total of 584 silent mutations in all three RVFV genome segments	[84]
	R566	A recombinant RVFV containing the small (S) segment of Clone 13 and the Large (L) and medium (M) segments of MP-12	[85]
	rRVF-ΔNSs:GFP-ΔNSm, ΔNSs-ΔNSm rRVFV	A recombinant RVFV ZH501 strain lacking nonstructural protein M (NSm) and NSs genes with and without the enhanced green fluorescent protein (eGFP) marker	[86,87,88,89]
	RVFV 4S	A recombinant RVFV having four segments, i.e., L, S without its NSs, and M split into a glycoprotein n (Gn) and a glycoprotein c (Gc) segment	[90,91,92,93,94]
DNA	RVFV + NSm DNA and RVFV-NSm DNA	DNA plasmid pWRG7077 encoding the RVFV M segment with or without the NSm gene	[95]
	RVFV cDNA N and RVFV cDNA GnGc	pcDNA3.1/V5-His^®^ TOPO (Invitrogen) encoding nucleoprotein N or Gn and Gc genes of RVFV	[96]
	Gn-cd3 DNA	DNA plasmid PTR600 expressing RVFV Gn coupled to 3 copies of the complement protein C3d as a molecular adjuvant	[97]
	pCMV-M4 and pCMV-N	pCMV vector (Clontech) encoding RVFV M segment (Nsm, Gn, and Gc) and N open reading frames (ORFs)	[98]
	pCMV-GnGc	pCMV vector (Clontech) encoding the MP-12 GnGc open reading frame starting from the fourth in-frame start codon	[99]
	peGn, pscDEC-eGn, and pscCD11c-eGn	pcDNA3.1 vector (Invitrogen) encoding the extracellular portion of RVFV Gn targeted to dendritic cells through fusion with a single-chain variable fragment (scFV) anti-ovine DEC205 (pscDEC-eGn) or scFV anti-ovine CD11c (pscCD11c-eGn) or untargeted (peGn)	[100]
	Gn-ELRs	A plasmid pCMVNSmGn encoding MP-12 NSm/Gn and various elastin-like recombinamers (ELRs)	[101]
	phRVF/Gn and phRVF/Gc	Biodegradable alginate (ALG)/poly-L-lysine (PLL) microcapsules entrapped with phMGFP (Promega) plasmids expressing Gn and Gc sequences of RVFV strain 1974-VNIIVViM and fusion protein F sequences of human parainfluenza virus 1 (HPIV-1)	[102]
Viral vectored	rLSDV–RVFV	Recombinant lumpy skin disease virus vaccine expressing RVFV Gn and Gc	[103,104]
	CAdVax-RVF	A nonreplicating complex adenovirus vector encoding RVFV ZH548M12 strain Gn and Gc sequences that were fused upstream with the human CD4 signal sequence	[105]
	NDFL-GnGC	A recombinant Newcastle disease virus LaSota strain containing a codon optimised RVFV Gn and Gc gene sequences from RVFV M35/74 strain	[106]
	vCOGnGc and vCOGnGcγ	Recombinant vaccinia virus attenuated by the deletion of IFN-γ binding protein gene and insertional inactivation of the thymidine kinase gene, expressing RVFV Gn and Gc from RVFV ZH548 strain with or without the human IFN-γ gene	[107]
	ChAdOx1-GnGC	A replication-deficient chimpanzee adenovirus vector ChAdOx1 encoding MP-12 Gn and Gc sequences	[108,109,110,111]
	rMVA-Gn/Gc and rMVA-N	Recombinant Modified Vaccinia Ankara (MVA) viruses encoding MP-12 protein N or Gn and Gc genes fused in-frame with human tissue plasminogen activator leader sequences at the N terminus	[99,112,113,114]
	MVA-GnGc-VP2, MVA-GnGc-NS1 and MVA-GnGc-NS1-Nt	Recombinant MVA expressing RVFV Gn and Gc sequences of MP-12 in addition to the Bluetongue virus (BTV) proteins VP2, NS1, or a truncated form of NS1 (NS1-Nt)	[115]
	rH_Gn-Gc	Equine herpesvirus type 1 strain RaCH expressing codon optimised RVFV Gn and Gc genes from the RVFV ZH501 strain	[116,117]
	rKS1/RVFV	A recombinant capripox virus expressing RVFV Gn and Gc glycoproteins	[117]
	rSRV9-eGn	An inactivated recombinant rabies virus vector rSRV9 cloned with a codon-optimised RVFV Gn ectodomain gene from MP-12	[118]
Subunit	Gne-S3	Gn ectodomain produced using the Drosophila expression system (Invitrogen, Carlsbad, CA, USA) and formulated in Stimune water-in-oil adjuvant (Prionics, Lelystad, The Netherlands)	[119]
	RVFV Gne and RVFV Gc	Baculovirus-expressed Gn ectodomain and Gc proteins formulated with a water-in-oil adjuvant montanide ISA25 (Seppic, France)	[120,121,122]
	Gn-_head_ MPSPs	RVFV Gn head attached to multimeric protein scaffold particles (MPSP) through spontaneous isopeptide bond formation between spytag and spycatcher “bacterial superglue” to yield antigen-decorated nanoparticles	[123]
Viral replicons	REP91-RVF(M) and Rgird-RVFV(M)	Alphavirus replicon vectors based on mosquito (AR86) and human (Girdwood) isolates of Sindbus virus expressing RVFV Gn, Gc, and NSm protein	[124]
	SINRepspGn and VEEVRepspGn	Sinbus and Venezuelan equine encephalitis virus TC-83 replicon particles expressing RVFV Gn	[125]
	NSR-Gn	A non-spreading RVFV Gn replicon containing RVFV L segment and an S segment encoding Gn in place of the NSs gene	[126]
	RVF-VRP	Single-cycle RVFV replicon particles rescued from BSR-T7/5 cells transfected with RVFV L, RVFS-ΔNSm/ΔNSs:GFP, and pC-GnGc plasmids	[127]
	scMP12 and scMP-12-mutNS	Single-cycle replicons rescued from BSR-T7/5 cells transfected with plasmids expressing MP-12 L, N, and Gn/Gc proteins, as well as L RNA, S RNA encoding N and GFP and M RNA encoding a mutant envelope protein lacking an endoplasmic reticulum retrieval signal. For scMP-12mutNs, the S RNA encoded N and NSsR16H/M250K	[128,129]
	RVFV RRP	Replicons made by transfecting baby hamster kidney cell lines maintaining L and S segments (whose NSs had been replaced with GFP) with the RVFV strain 35/74 M genome segment starting at the fourth in-frame start codon	[130]
Virus-like particles	Ren-VLPs	VLPs made by transfecting 293T cells with the expression plasmids p18 subcloned with L, N and M segments and a reporter plasmid containing Renilla luciferase gene (Ren-Luc)	[131]
	RVFV chimVLPs	Chimeric VLPs containing RVFV Gn and Gc, nucleoprotein N, and the gag protein of Moloney murine leukemia virus	[132]
	RVFV GnGc VLP	VLPs generated through the expression of Gn and Gc in the Drosophila insect cell expression system	[133]

## Data Availability

Not applicable.

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
