# Peer review of "An Overview of Rift Valley Fever Vaccine Development Strategies"

_vaccines, 2022, doi:10.3390/vaccines10111794_

Round 1
Reviewer 1 Report
This is a well-written review on RVFV vaccines. The manuscript is well organized and appropriate references used. The authors did discuss the need for DIVA vaccines, which is an important topic in vaccine development. Well done!
Author Response
Dear Reviewer, in response to your comment, please find attached a revised manuscript with track changes. The English language has been proofread and the necessary corrections made. Thank you.

Reviewer 2 Report
Excellent manuscript. No comments.
Author Response
Dear Reviewer, I have made minor English proofreading checks to the manuscript in response to a comment I received from two of the reviewers who checked the box "English language and style are fine/minor spell check required". Please find attached the revised manuscript with track changes. Thank you

Reviewer 3 Report
In this manuscript, the authors explore the progress in Rift Valley Fever vaccine research and development. This is an exceptionally well-written, exhaustive review of an array of strategies in RVF vaccine development. Table 1 is very clear and concise. I recommend this manuscript for publication in Vaccines in present form.
Author Response

(The authors gave the same response as above.)
